# A Novel Simple Fabrication Method for Mechanically Robust Superhydrophobic 2024 Aluminum Alloy Surfaces

**Li-Mei Shan \*, Guo-Biao Liu, Hua Tang, Zhi-Hong Li and Ju-Ying Wu**

Department of Materials Science and Engineering, Sichuan Engineering Technical College, Deyang 618000, China
\* Correspondence: slm@scetc.edu.cn; Tel.: +86-838-2658626

**Abstract:** The mechanical durability of a superhydrophobic aluminum alloy surface is an important indicator of its practical use. Herein, we propose a strategy to prepare a superhydrophobic 2024 aluminum alloy surface with highly enhanced mechanical durability by using a two-step chemical etching method, using a NaOH solution as the etchant in step one and a $Na_2CO_3$ solution as the etchant in step two. Robust mechanical durability was studied by static contact angle tests before and after an abrasion test, potentiodynamic polarization measurements after an abrasion test and electrochemical impedance spectroscopy tests after an abrasion test. Furthermore, the mechanism for enhanced mechanical durability was investigated through scanning of electron microscopy images, energy-dispersive X-ray spectra, Fourier transform infrared spectra and X-ray photoelectron spectra. The testing results indicate that a hierarchical rough surface consisting of regular micro-scale dents and some nano-scale fibers in the micro-scale dents, obtained with the two-step chemical etching method, contributes to highly enhanced mechanical durability. Meanwhile, the as-prepared superhydrophobic 2024 aluminum alloy surface retained a silvery color instead of the black shown on the superhydrophobic 2024 aluminum alloy surface prepared by a conventional one-step chemical etching method using NaOH solution as the etchant.

**Keywords:** superhydrophobic; mechanical durability; 2024 aluminum alloy; two-step chemical etching method

## 1. Introduction

Due to the advantages of low specific weight, high specific strength, excellent thermal conductivity and low cost, aluminum alloys have been widely applied in many fields, such as architecture, traffic, industry and household items [1]. However, aluminum alloys are easily corroded in environments containing chloride ions (for example, seawater) because their corrosion resistance becomes poor in those environments [2]. To enhance corrosion resistance of aluminum alloys, many types of surface modification methods based on aluminum-alloy substrates have been developed [3]. Preparation of superhydrophobic surfaces on aluminum-alloy substrates is a promising surface modification method to enhance corrosion resistance of aluminum alloys because those superhydrophobic surfaces can minimize the contact area between the corrosive solution and the surface of aluminum alloys [4,5]. However, it is difficult to obtain a mechanically robust superhydrophobic surface based on aluminum-alloy substrates. Therefore, research about enhancing mechanical durability of superhydrophobic aluminum alloy surfaces attracts great attention [5–15].

In general, a two-step methodology including fabrication of a rough surface consisting of micro/nano structures and chemical modification using some low-surface-energy materials has been widely used to create superhydrophobic aluminum alloy surfaces [4,5,16]. Because a rough surface is critical to superhydrophobicity and mechanical durability of superhydrophobic aluminum alloy surfaces, many approaches, such as laser texture [6,7,17–29], boiling-water treatment [30–32], electrodeposition [8,33–35], a combination of micromilling and grinding [36], hydrothermal synthesis [9,10], spin coating [37], anodizing [38–40], a

combination of white-light interferometry and chemical etching [11], a combination of laser texture and boiling-water treatment [41], a combination of boiling-water treatment and chemical etching [42] and chemical etching by itself [12–15,43–63], have been developed to construct rough surfaces on aluminum-alloy substrates. Among these approaches, chemical etching has been highlighted on account of its low cost and high efficiency. Furthermore, several kinds of chemicals, including acid [12–14,43–49,52,53], salt [50,51] and alkali [15,54–64] solutions, have been widely used as etchants. Among these etchants, alkali solution is preferable because it is more beneficial for the environment.

Due to its low cost, NaOH solution is widely used as the alkali chemical etchant for preparing superhydrophobic aluminum alloy surfaces. For example, Guo et al. prepared superhydrophobic 2024 aluminum alloy surfaces by using 1 mol $L^{-1}$ NaOH solution as the chemical etchant and perfluorononane ($C_9F_{20}$) and poly (dimethylsiloxane) vinyl terminated (PDMSVT) as low-surface-energy materials for the first time [60]. Yang et al. made a full investigation of influence of NaOH-solution etching temperature on morphologies and superhydrophobicities of superhydrophobic 3003 aluminum alloy surfaces. They drew a conclusion that a high etching temperature of NaOH solution is necessary to fabricate a rough surface with micro/nano structures and a superhydrophobic 3003 aluminum alloy surface [62]. Cho et al. prepared superhydrophobic pure aluminum alloy surfaces by using 0.05 mol $L^{-1}$ NaOH solution at 80 °C as the chemical etchant [15]. As-prepared superhydrophobic pure aluminum alloy surfaces showed relatively robust mechanical durability. Nevertheless, in our preliminary experiment, we found that the superhydrophobic surface on a 2024 aluminum-alloy substrate prepared by the one-step chemical etching method using NaOH solution as the etchant had two shortcomings. First, the superhydrophobic surface on 2024 aluminum-alloy substrate prepared by the one-step chemical etching method using NaOH solution as the etchant presented an undesirable black color. In addition, the as-prepared black superhydrophobic surface on the 2024 aluminum-alloy substrate showed poor mechanical durability. In this work, to prepare a mechanically robust non-black superhydrophobic surface on 2024 aluminum-alloy substrate, a two-step chemical etching method using a NaOH solution as the etchant in step one and a $Na_2CO_3$ solution as the etchant in step two is presented for the first time. Meanwhile, the chemical etching mechanism of the two-step chemical etching method is proposed. Furthermore, the reasons for improved mechanical durability of the as-prepared non-black superhydrophobic 2024 aluminum alloy surface were investigated.

## 2. Materials and Methods

### 2.1. Materials

The necessary 2024 aluminum alloy sheets with a size of $20 \times 20 \times 4$ mm$^3$ each were purchased from Shanghai Yunxian Metal Materials Co. Ltd. (Shanghai, China). and used as substrates. The chemical composition is shown in Table 1. Sodium hydroxide (NaOH), sodium carbonate ($Na_2CO_3$), hydrochloric acid (HCl), stearic acid ($C_{18}H_{36}O_2$), acetone ($CH_3COCH_3$) and ethanol ($C_2H_5OH$) were purchased from Chengdu Kelong Chemical Co. Ltd (Chengdu, China). These chemical reagents were analytical grade.

**Table 1.** Chemical composition of the purchased 2024 aluminum alloy sheet.

| Elements | Cu | Si | Fe | Mn | Mg | Cr | Zn | Ti | Al |
|---|---|---|---|---|---|---|---|---|---|
| wt% | 4.80 | 0.09 | 0.27 | 0.64 | 1.61 | 0.01 | 0.13 | 0.03 | 92.42 |

### 2.2. Preparation of the Superhydrophobic 2024 Aluminum Alloy Surface

Superhydrophobic 2024 aluminum alloy surfaces were acquired using a two-step methodology including fabrication of a rough surface using a two-step chemical etching method and chemical modification with stearic acid. A schematic diagram of preparation of a superhydrophobic 2024 aluminum alloy surface is shown in Figure 1. First, a pre-treatment was conducted. The 2024 aluminum alloy sheets were polished using SiC

abrasive papers of 600# and 1000#. The polished sheets were ultrasonically cleaned using ethanol and degreased using acetone. Second, a two-step chemical etching method was carried out to create a rough surface with hierarchical micro/nano structures. In order to create micrometer scale structures on the 2024 aluminum alloy surface, the degreasedsheets were immersed into a NaOH solution of 1 mol $L^{-1}$ for 30 min at 25 °C. Then, the sheets were washed using 1 mol $L^{-1}$ HCl solution and deionized water, respectively. To create nano structures, the primary etched sheets were immersed into a $Na_2CO_3$ solution of 0.1 mol $L^{-1}$ for 2 min at 25 °C. Finally, chemical modification of sheets with rough surfaces was conducted. The etched sheets were immersed in an ethanol solution with 0.05 mol $L^{-1}$ stearic acid for 30 min at 25 °C. Afterwards, the chemical modified sheets were dried at 80 °C in an oven.

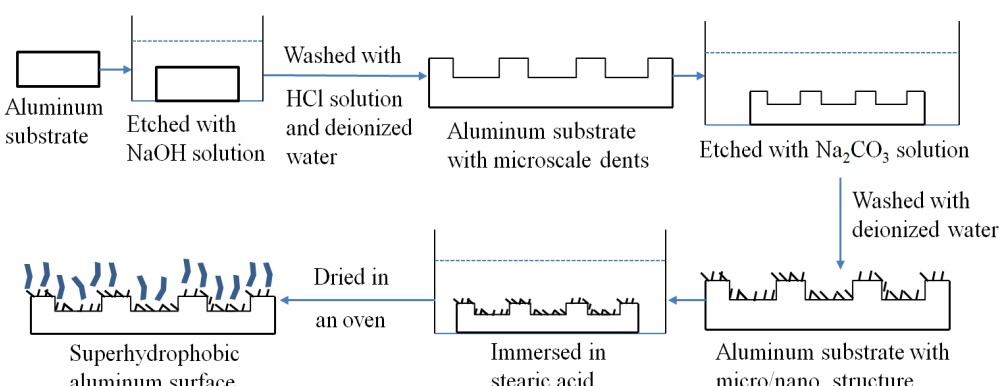

**Figure 1.** Schematic diagram for preparing superhydrophobic 2024 aluminum alloy surface.

For comparison, another type of superhydrophobic 2024 aluminum alloy surface was obtained using a two-step method including fabrication of a rough surface using a one-step chemical etching method and chemical modification with stearic acid. First, a pre-treatment was conducted. The 2024 aluminum alloy sheets were polished using SiC abrasive papers of 600# and 1000#. The polished sheets were ultrasonically cleaned using ethanol and degreased using acetone. Second, a one-step chemical etching method was carried out to create rough surfaces. The degreased sheets were immersed into a NaOH solution of 0.25 mol $L^{-1}$ for 5 min at 90 °C. Then, the sheets were washed using deionized water. Finally, chemical modification of sheets with rough surfaces was conducted. The etched sheets were immersed into an ethanol solution with 0.05 mol $L^{-1}$ stearic acid for 30 min at 25 °C. Afterwards, the chemical modified sheets were dried at 80 °C in an oven.

*2.3. Characterisation of As-Prepared Superhydrophobic Surface*

Water contact angles were obtained through the sessile drop method using a contact angle goniometer (DSA100, Kruss, Germany). Morphologies of samples were observed on a scanning electron microscope (Gemini SEM 300, Zeiss, Oberkochen, Germany). Compositions of samples were studied using energy-dispersive X-ray (EDX) spectra obtained with an EDX spectrometer (Aztec X-MAX80, Oxford, UK), Fourier transform infrared (FTIR) spectra obtained with an FTIR spectrophotometer (Nicolet Is5, Thermo Fisher Scientific, Waltham, MA, USA) in the range 500 to 4000 $cm^{-1}$ and X-ray photoelectron spectra obtained with an X-ray photoelectron spectrometer (Escalab 250Xi, Thermo Fisher Scientific, Waltham, MA, USA). Overview spectra (from 0 to 1350 eV) were recorded using Al K$\alpha$ radiation (hv = 1486.6 eV) and binding energies were calibrated based on 284.8 eV.

Abrasion tests of as-prepared samples were performed with the following steps: The testing sample was placed on a SiC abrasive paper of 800#. The superhydrophobic surface of the testing sample faced the abrasive paper. Meanwhile, an iron block was loaded on the testing sample. The testing sample, with a load of 0.03 kg $cm^{-2}$, was moved in one direction with 5 mm $s^{-1}$ at two strokes of 15 cm.

Potentiodynamic polarization measurements were carried out on an electrochemical workstation (CHI660D, Shanghai CH Instruments, Shanghai, China) using a three-electrode system at room temperature. The as-prepared sample was used as a working electrode. A platinum sheet was used as a counter electrode. An Ag/AgCl/saturated KCl electrode was used as a reference electrode. To realize stable open-circuit potential before collection of potentiodynamic polarization data, three electrodes were immersed in 3.5 wt% NaCl solution for 30 min. Potentiodynamic polarization curves were achieved at stable open-circuit potential and at a scanning rate of 1 mV s$^{-1}$. Electrochemical impedance spectroscopy (EIS) tests were also carried out on the CHI660D electrochemical workstation. EIS curves were acquired at frequencies in a range of 100 kHz to 10 mHz.

## 3. Results and Discussion

Because water repellency was not the main focus in this study, only static water contact angles were obtained. SEM images and static water contact angles of the samples treated at different conditions are demonstrated in Figure 2a–g,a′–g′. Figure 2a–e demonstrates morphologies and contact angles of samples treated at various conditions using the two-step chemical etching method. As shown in Figure 2b,b′, the NaOH-etched sample at 25 °C presented a rough surface with irregular micro-scale particles, cracks and irregular nano-scale fibers on the surface of the micro-scale particles, which is in accord with the results in the previous literature [60,62]. Furthermore, as shown in the inset of Figure 2b, the NaOH-etched sample at 25 °C showed a black surface, which was not observed in other types of NaOH-solution-etched aluminum alloy surfaces or documented in the previous literature [15,54–64]. As shown in Figure 2c, the NaOH-etched aluminum alloy at 25 °C after HCl washing presented a relatively regular rough surface with regular micro-scale dents. The inset of Figure 2c presents a silvery white surface, indicating removal of the black substance. As shown in Figure 2d,d′, after $Na_2CO_3$-etching, the $Na_2CO_3$-etched sample presented a relatively regular rough surface with regular micro-scale dents and some nano-scale fibers in the micro-scale dents. The inset of Figure 2d also shows a silvery white surface. As shown in Figure 2e,e′, the stearic-acid-treated sample presented a similar, relatively regular rough surface with regular micro-scale dents. The nano-scale fibers in Figure 2d′ turned into nano-scale grids. The inset of Figure 2e also shows a silvery white surface. Figure 2f,f′,g,g′ demonstrates the morphologies of the samples prepared using the one-step chemical etching method. As shown in Figure 2f,g, samples prepared using the one-step chemical etching method presented a rough surface with irregular micro-scale particles and cracks, which is in accord with the results in the previous literature [57,60]. However, as shown in the insets of Figure 2f,g, the samples showed a black surface: a result that was not documented in the previous literature [15,54–64]. As shown in the insets of Figure 2e,e′,g,g′ and Table 2, both stearic-acid-treated samples had good superhydrophobicity, with a contact angle of over 160°. The regular rough surface of the sample prepared using the two-step chemical etching method was expected to facilitate robust mechanical durability for the as-prepared superhydrophobic aluminum alloy surface.

**Table 2.** Static water contact angles for (a) untreated aluminum alloy, (b) NaOH-etched aluminum alloy at 25 °C after HCl washing, (c) HCl-washed aluminum alloy after $Na_2CO_3$ etching, (d) $Na_2CO_3$-etched aluminum alloy at 25 °C, (e) stearic-acid-treated aluminum alloy after $Na_2CO_3$ etching, (f) NaOH-etched aluminum alloy at 90 °C, (g) stearic-acid-treated aluminum alloy after NaOH etching at 90 °C, (h) stearic-acid-treated sample after $Na_2CO_3$ etching after abrasion test and (i) stearic-acid-treated sample after NaOH etching at 90 °C after abrasion test.

| Samples | (a) | (b) | (c) | (d) | (e) | (f) | (g) | (h) | (i) |
|---|---|---|---|---|---|---|---|---|---|
| SWCA *(°) | 60.5 | 21.8 | 108.7 | 108.1 | 161.7 | 58.9 | 164.7 | 144.6 | 133.9 |

* SWCA means static water contact angle.

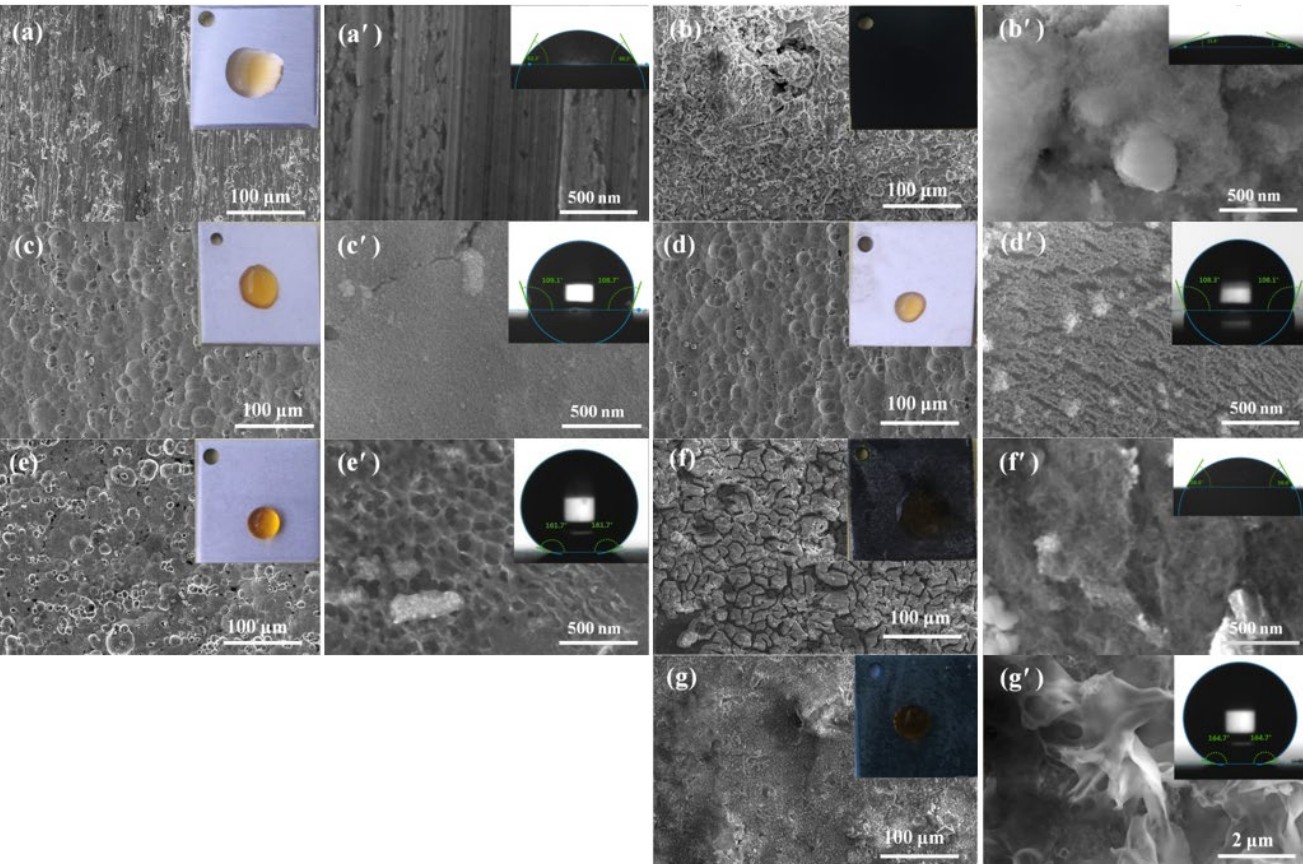

**Figure 2.** SEM images of (**a**,**a′**) untreated aluminum alloy, (**b**,**b′**) NaOH-etched aluminum alloy at 25 °C after HCl washing, (**c**,**c′**) HCl-washed aluminum alloy after Na₂CO₃ etching, (**d**,**d′**) Na₂CO₃-etched aluminum alloy at 25 °C, (**e**,**e′**) stearic-acid-treated aluminum alloy after Na₂CO₃ etching, (**f**,**f′**) NaOH-etched aluminum alloy at 90 °C and (**g**,**g′**) stearic-acid-treated aluminum alloy after NaOH etching at 90 °C. The inset is the image of the contact angle.

The compositions of the samples treated at different conditions were characterized by conduction of EDX measurements. As shown in Figure 3, along with the Cu, Mn, Mg, Fe and Si elements, the O and C element were present on the stearic-acid-treated sample after Na₂CO₃-etching. The presence of the C element can be ascribed to chemical modification with stearic acid. Table 3 exhibits chemical compositions of samples treated at different conditions. The content of the O element for the Na₂CO₃-etched sample afterNaOH etching was higher than that of the pristine sample, indicating oxidation of some metal elements. The conclusion was further verified by the following XPS analyses.

XPS spectra were obtained to further study chemical compositions of samples treated at different conditions. Figure 4a shows the survey of XPS spectra. In XPS spectra of the stearic-acid-treated aluminum alloy after Na₂CO₃ etching and of the stearic-acid-treated aluminum alloy after NaOH etching at 90 °C, characteristic peaks of Al and Cu for both stearic-acid-treated samples are clearly present, indicating a thin superhydrophobic film for both stearic-acid-treated samples. High-resolution spectra of the Cu and Al elements are shown in Figure 4b,c. In Figure 4b, the 933.6 eV peaks for the untreated sample and the stearic-acid-treated sample after Na₂CO₃ etching can mainly be ascribed to 2p 3/2 peaks of Cu, which was deduced based on the results in the previous literature [64] and the silvery color of the samples. The 934.8 eV peak of the stearic-acid-treated sample after NaOH etching at 90 °C can mainly be ascribed to 2p 3/2 peaks of CuO, which was deduced based on the results in the previous literature [65] and the black color of the sample. The result of the high-resolution XPS spectra of Cu is in agreement with that of the chemical composition change detected by EDX mapping. In Figure 4c, the 72.8 eV peak of the untreated sample

can be ascribed to 2p 3/2 peaks of Al, and the 74.8 eV peak can be ascribed to 2p 3/2 peaks of $Al_2O_3$ [65]. The 72.8 eV peak of the stearic-acid-treated sample after $Na_2CO_3$ etching can be ascribed to 2p 3/2 peaks of Al, and the 74.4 eV peak can be ascribed to 2p 3/2 peaks of $Al(OH)_3$. The 74.4 eV peak of the stearic-acid-treated sample after NaOH etching at 90 °C can be ascribed to 2p 3/2 peak of $Al(OH)_3$ [55]. The result of the high-resolution XPS spectra of Al is also in agreement with that of the chemical composition change detected by EDX mapping.

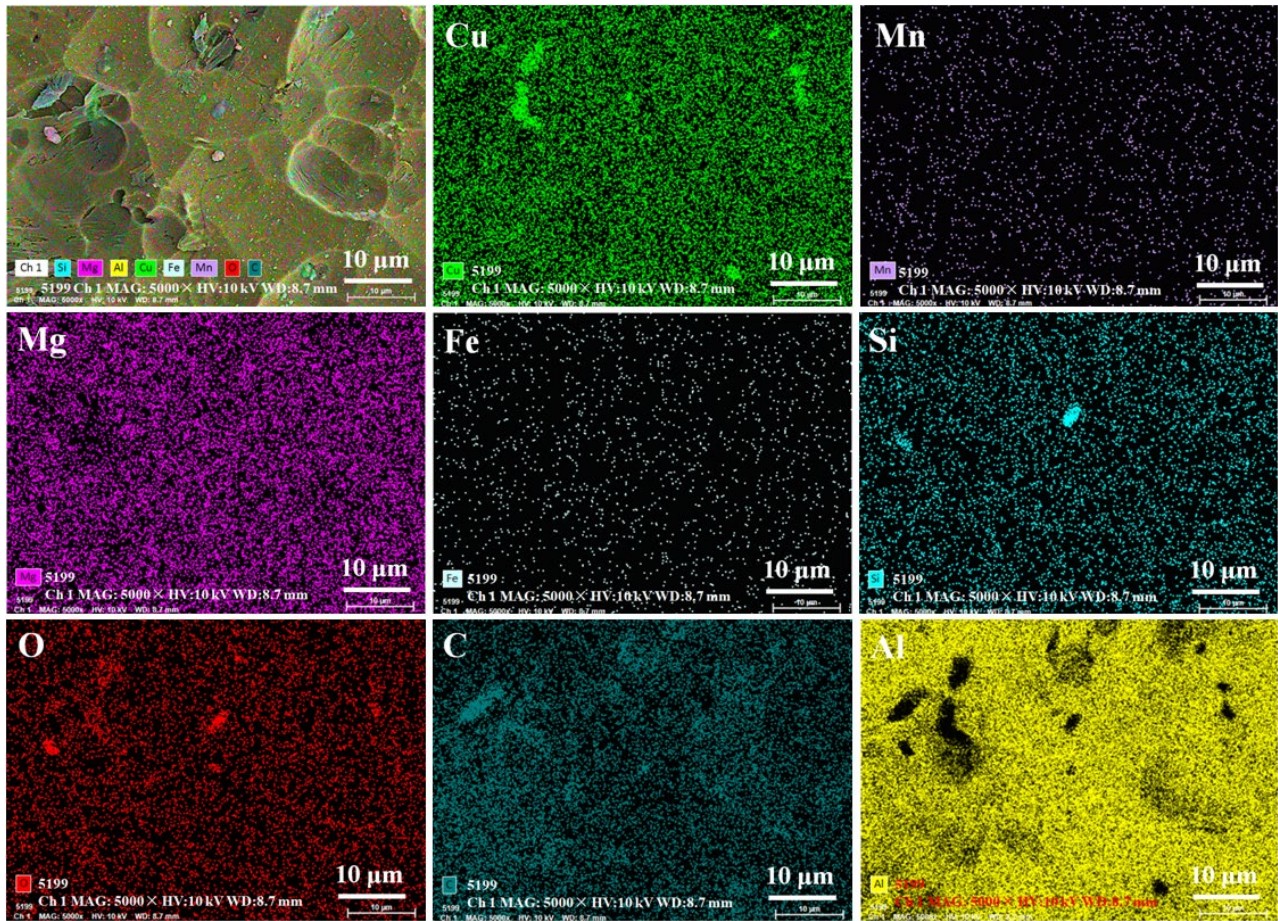

**Figure 3.** EDX mapping images of the stearic-acid-treated sample after $Na_2CO_3$-etching.

**Table 3.** Chemical compositions of (a) untreated aluminum alloy, (b) NaOH-etched aluminum alloy at 25 °C after HCl washing, (c) HCl-washed aluminum alloy after $Na_2CO_3$ etching, (d) $Na_2CO_3$-etched aluminum alloy at 25 °C, (e) stearic-acid-treated aluminum alloy after $Na_2CO_3$ etching, (f) NaOH-etched aluminum alloy at 90 °C and (g) stearic-acid-treated aluminum alloy after NaOH etching at 90 °C.

| Samples | Cu wt% | Mn wt% | Mg wt% | Fe wt% | Si wt% | O wt% | C wt% | Al wt% |
|---------|--------|--------|--------|--------|--------|-------|-------|--------|
| (a) | 4.63 | 0.82 | 1.19 | 0.46 | 0.46 | 1.47 | - | 90.98 |
| (b) | 35.97 | 3.82 | 11.55 | 0.68 | 1.93 | 37.30 | - | 8.76 |
| (c) | 5.74 | 0.32 | 1.64 | 0.01 | 0.34 | 3.01 | - | 88.95 |
| (d) | 8.29 | 0.46 | 1.55 | 0.79 | 0.08 | 5.11 | - | 83.72 |
| (e) | 4.87 | 0.30 | 1.23 | 0.06 | 0.42 | 2.39 | 21.36 | 69.36 |
| (f) | 31.33 | 5.15 | 11.35 | 0.77 | 0.44 | 35.56 | - | 15.42 |
| (g) | 27.31 | 2.87 | 9.06 | 0.80 | 4.41 | 35.84 | 8.22 | 11.48 |

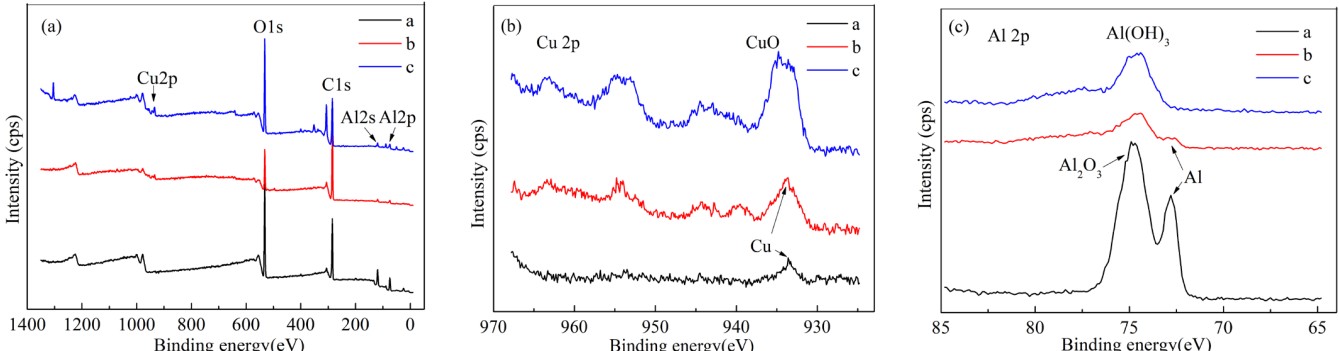

**Figure 4.** XPS spectra of (**a**) untreated aluminum alloy, (**b**) stearic-acid-treated aluminum alloy after $Na_2CO_3$ etching and (**c**) stearic-acid-treated aluminum alloy after NaOH-etching at 90 °C.

Deduced from analyses of SEM images, SEM-EDX mapping and XPS spectra, the chemical etching mechanism of the two-step chemical etching method can be described as the following equations. The NaOH solution etching process can be described by Equations (2)–(6). The HCl solution washing process can be described by Equations (7) and (8). The $Na_2CO_3$ solution etching process can be described by Equations (9)–(11). Due to the generation of $CO_2$ in the $Na_2CO_3$ solution etching process, the reaction between copper and NaOH was restrained. Therefore, the $Na_2CO_3$-etched aluminum alloy and the stearic-acid-treated aluminum alloy after $Na_2CO_3$ etching still showed a silvery white surface.

$$2Al + 2NaOH + 2H_2O \rightarrow 2NaAlO_2 + 3H_2 \tag{1}$$

$$NaAlO_2 + H_2O \rightarrow NaOH + Al(OH)_3 \tag{2}$$

$$2Al(OH)_3 \rightarrow Al_2O_3 + 3H_2O \tag{3}$$

$$2Cu + 4NaOH + O_2 \rightarrow 2Na_2CuO_2 + 2H_2O \tag{4}$$

$$NaCuO_2 + 2H_2O \rightarrow 2NaOH + Cu(OH)_2 \tag{5}$$

$$2Cu(OH)_2 \rightarrow 2CuO + 2H_2O \tag{6}$$

$$Al_2O_3 + 6HCl \rightarrow 2AlCl_3 + 3H_2O \tag{7}$$

$$CuO + 2HCl \rightarrow 2CuCl_2 + H_2O \tag{8}$$

$$2Al + Na_2CO_3 + 3H_2O \rightarrow 2NaAlO_2 + 3H_2 + CO_2 \tag{9}$$

$$NaAlO_2 + H_2O \rightarrow NaOH + Al(OH)_3 \tag{10}$$

$$2Al(OH)_3 \rightarrow Al_2O_3 + 3H_2O \tag{11}$$

FTIR spectra were obtained to investigate the chemical functional group and chemical structure of samples treated at different conditions. Figure 5 shows the FTIR spectra of the untreated aluminum alloy, the stearic-acid-treated aluminum alloy after $Na_2CO_3$ etching and the stearic-acid-treated aluminum alloy after NaOH etching at 90 °C. The peaks in the FTIR spectrum of the stearic-acid-treated sample after $Na_2CO_3$ etching are much weaker than those in the FTIR spectrum of the stearic-acid-treated sample after NaOH etching at 90 °C, which indicates a thinner superhydrophobic film on the stearic-acid-treated sample after $Na_2CO_3$ etching. This result is in accord with that observed in the SEM images in Figure 2e,e',g,g'. The broad peak at 3400 $cm^{-1}$ can be ascribed to the –OH band of $Al(OH)_3$ [54]. The peaks at 2915 and 2847 $cm^{-1}$ are attributed to the C–H and –$CH_2$ bands [58,62], respectively. The peak at 1586 $cm^{-1}$ is attributed to the –COOAl band [58]. The presence of –OH, C–H, –$CH_2$ and –COOAl bands verifies successful chemical modification with stearic acid on etched aluminum-alloy substrates. The peaks at 1410 and 1360 $cm^{-1}$ are attributed to the –AlO band [62]. Furthermore, the peaks at 717, 554 and 405 $cm^{-1}$ are attributed to the Al–O band [58,62]. The weak peaks at 3400, 717, 554

and 405 cm$^{-1}$ for stearic-acid-treated aluminum alloy after Na$_2$CO$_3$ etching indicate a thin Al(OH)$_3$ film, which is also observed in the SEM image in Figure 2d,d'.

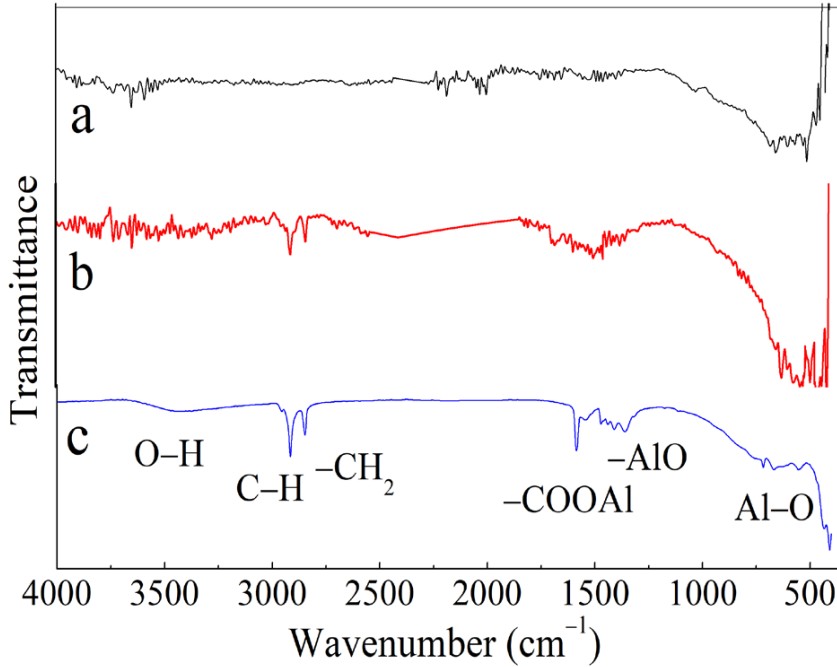

**Figure 5.** FTIR spectra of (a) untreated aluminum alloy, (b) stearic-acid-treated aluminum alloy after Na$_2$CO$_3$ etching and (c) stearic-acid-treated aluminum alloy after NaOH etching at 90 °C.

Digital images of surfaces on stearic-acid-treated samples after Na$_2$CO$_3$ etching and stearic-acid-treated samples after NaOH etching at 90 °C before and after abrasion tests were obtained to investigate mechanical durability. As shown in Figure 6a,b, the stearic-acid-treated sample after Na$_2$CO$_3$ etching presented a silvery color, and the stearic-acid-treated sample after Na$_2$CO$_3$ etching after the abrasion test showed a similar color. However, as shown in Figure 6c,d, the stearic-acid-treated sample after NaOH etching at 90 °C presented a black color and the stearic-acid-treated sample after NaOH etching at 90 °C after the abrasion test showed a light gray color. Figure 6e,f show SEM images of surfaces on the stearic-acid-treated sample after Na$_2$CO$_3$-etching after the abrasion test and on the stearic-acid-treated sample after NaOH etching at 90 °C after the abrasion test. As shown in Figure 6e,f, the stearic-acid-treated sample after NaOH etching at 90 °C after the abrasion test showed a severely broken surface. As listed in Table 2, the static contact angle changed from 164.7° to 133.9° after the abrasion test. Variation of the water contact angle for the stearic-acid-treated sample after NaOH etching at 90 °C was 30.8°. In contrast to the severely broken surface of the stearic-acid-treated sample after NaOH etching at 90 °C after the abrasion test, the stearic-acid-treated sample after Na$_2$CO$_3$ etching after the abrasion test showed a relatively slightly broken surface, which consisted of some slight broken micro-scale dents and intact nano-scale fibers in the micro-scale dents. As listed in Table 2, the static contact angle changed from 161.7° to 144.6° after the abrasion test. Variation of the water contact angle for the stearic-acid-treated sample after Na$_2$CO$_3$ etching was 17.1°, which is bigger than the 10.7° variation of water contact angle for superhydrophobic surfaces on 1060 aluminum-alloy substrates in the previous literature [66]. This can be ascribed to the different aluminum-alloy substrates. In contrast to the severely broken surface and big variation of the static contact angle for the stearic-acid-treated sample after NaOH etching at 90 °C, the slightly broken surface and smaller variation of the static contact angle for the stearic-acid-treated sample after Na$_2$CO$_3$ etching indicated highly improved mechanical durability of the stearic-acid-treated sample after Na$_2$CO$_3$ etching.

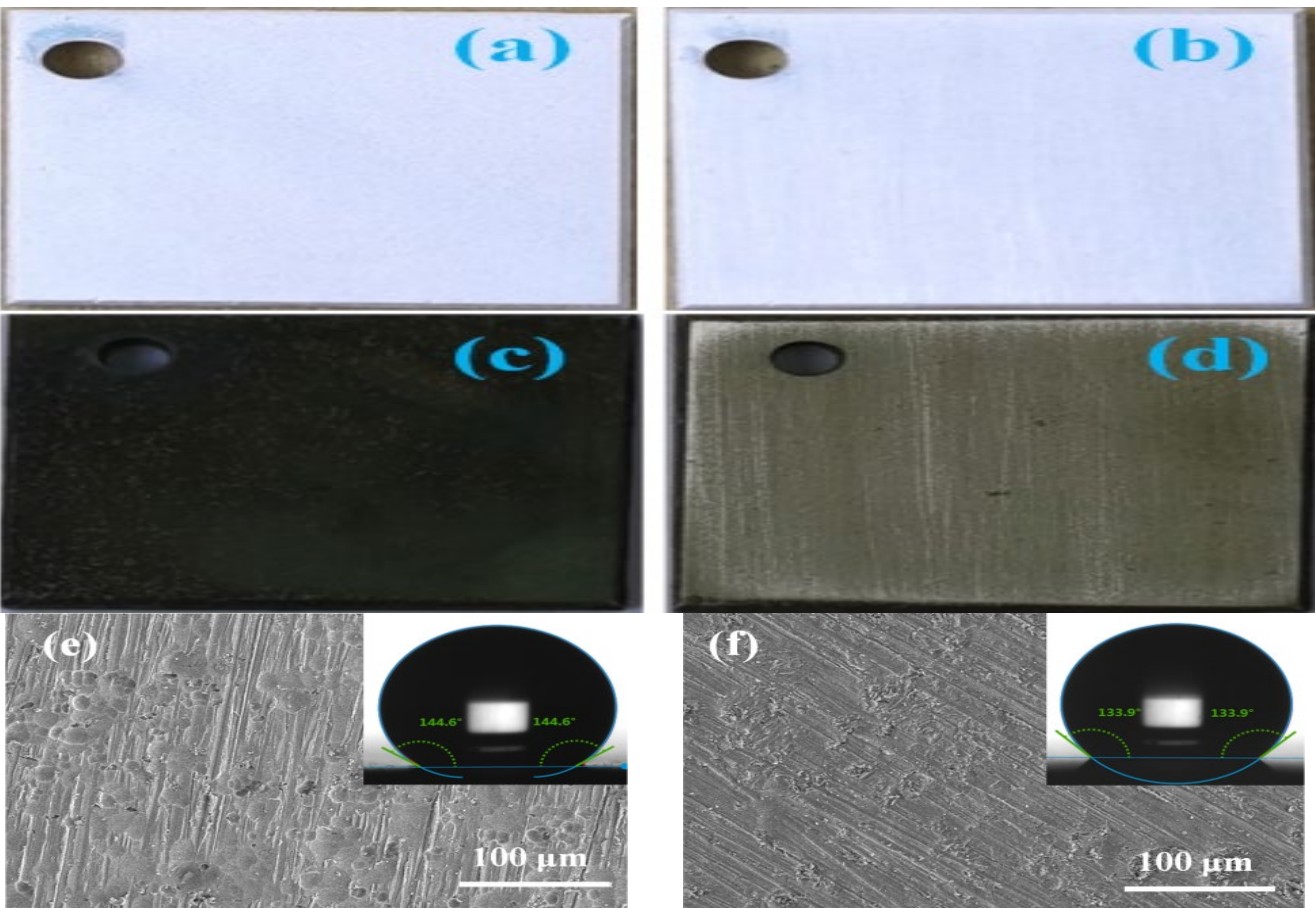

**Figure 6.** Digital images of (**a**) stearic-acid-treated sample after $Na_2CO_3$ etching, (**b**) stearic-acid-treated sample after $Na_2CO_3$ etching after abrasion test, (**c**) stearic-acid-treated sample after NaOH etching at 90 °C and (**d**) stearic-acid-treated sample after NaOH etching at 90 °C after abrasion test. SEM images of (**e**) stearic-acid-treated sample after $Na_2CO_3$ etching after abrasion test, (**f**) stearic-acid-treated sample after NaOH etching at 90 °C after abrasion test. The inset is the image of the contact angle.

To further confirm highly improved mechanical durability of the stearic-acid-treated sample after $Na_2CO_3$ etching, the anti-corrosive properties of the stearic-acid-treated sample after $Na_2CO_3$-etching after the abrasion test and of the stearic-acid-treated sample after NaOH etching at 90 °C after the abrasion test were investigated via potentiodynamic polarization curves as well as corrosion current density ($i_{corr}$), corrosion potential ($E_{corr}$) and polarization resistance ($R_p$). Potentiodynamic polarization curves are shown in Figure 7. Corresponding data are listed in Table 4. As listed in Table 4, the values of $i_{corr}$ and $R_p$ for the stearic-acid-treated sample after $Na_2CO_3$ etching after the abrasion test were 0.15 μA cm$^{-2}$ and 268.7 kΩ cm$^2$, respectively. However, the values of $i_{corr}$ and $R_p$ for the stearic-acid-treated sample after NaOH etching at 90 °C after the abrasion test were 0.68 μA cm$^{-2}$ and 35.2 kΩ cm$^2$, respectively. The smaller $i_{corr}$ value and bigger $R_p$ value for the stearic-acid-treated sample after $Na_2CO_3$ etching after the abrasion test indicate better corrosion resistance, which verifies highly improved mechanical durability of the stearic-acid-treated sample after $Na_2CO_3$ etching.

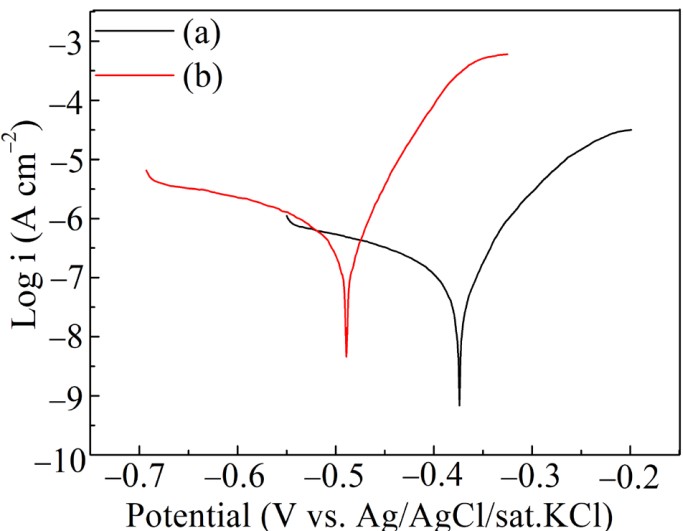

**Figure 7.** Potentiodynamic polarization curves of (a) stearic-acid-treated sample after $Na_2CO_3$ etching after abrasion test and (b) stearic-acid-treated sample after NaOH etching at 90 °C after abrasion test.

**Table 4.** Corrosion current density ($i_{corr}$), polarization resistance ($R_p$) and corrosion potential ($E_{corr}$) of (a) stearic-acid-treated sample after $Na_2CO_3$ etching after abrasion test and (b) stearic-acid-treated sample after NaOH etching at 90 °C after abrasion test.

| Sample | $E_{corr}$ (mV vs. Ag/AgCl/sat. KCl) | $i_{corr}$ ($\mu$A cm$^{-2}$) | $R_p$ (k$\Omega$ cm$^2$) |
|--------|--------|--------|--------|
| (a) | 382 | 0.15 | 268.7 |
| (b) | 490 | 0.68 | 35.2 |

To further confirm highly improved mechanical durability of the stearic-acid-treated sample after $Na_2CO_3$ etching, EIS measurements were taken to study the anti-corrosive properties of the stearic-acid-treated sample after $Na_2CO_3$-etching after the abrasion test and of the stearic-acid-treated sample after NaOH etching at 90 °C after the abrasion test. Nyquist plots, presenting two semicircles, are shown in Figure 8a. The smaller semicircle at high frequency represents resistance of the superhydrophobic thin films ($R_{SH}$). The second large semicircle represents charge-transfer resistance ($R_{ctSH}$) of the double layer at the interface between the superhydrophobic surface and the salt solution. The Nyquist plots can be fitted to the equivalent electrical circuit with two *CPEs* shown in Figure 8b [58,62]. In the equivalent electrical circuit, $R_s$ is resistance of the solution. $R_{SH}$ represents resistance of the superhydrophobic thin films. $CPE_{SH}$ is the constant phase element associated with the dielectric nature of the superhydrophobic film. $R_{ctSH}$ represents charge-transfer resistance of the double layer at the interface between the superhydrophobic surface and the salt solution. $CPE_{ctSH}$ is the constant phase element associated with the double layer at the interface between the superhydrophobic film surface and the salt solution. The $R_{SH}$ and $R_{ctSH}$ values of the stearic-acid-treated sample after $Na_2CO_3$ etching after the abrasion test were 28.32 and 112.06 k$\Omega$ cm$^2$, respectively. Nevertheless, the $R_{SH}$ and $R_{ctSH}$ values of the stearic-acid-treated sample after NaOH etching at 90 °C after the abrasion test were 12.77 and 45.49 k$\Omega$ cm$^2$, respectively. The much larger $R_{SH}$ and $R_{ctSH}$ values of the stearic-acid-treated sample after $Na_2CO_3$ etching after the abrasion test indicate a better anti-corrosive property. This result is in agreement with that obtained from potentiodynamic polarization measurements and is shown in Figure 7 and listed in Table 4.

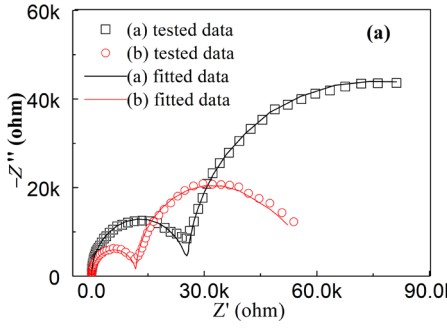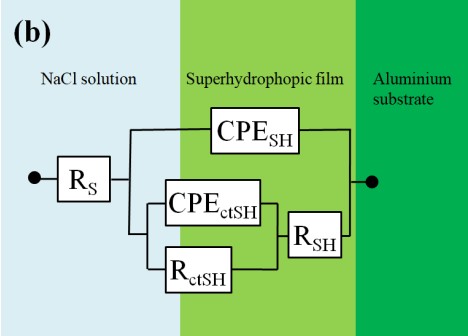

**Figure 8.** EIS spectra and electrical equivalent circuit of (**a**) stearic-acid-treated sample after Na₂CO₃ etching after abrasion test and (**b**) stearic-acid-treated sample after NaOH etching at 90 °C after abrasion test.

## 4. Conclusions

In this work, amechanically robust non-black superhydrophobic 2024 aluminum alloy surface was successfully prepared by a two-step chemical etching method, using NaOH solution as the etchant of step one and Na₂CO₃ solution as the etchant of step two. The variation of static contact angles for the as-prepared non-black superhydrophobic 2024 aluminum alloy surface before and after abrasion test was 17.1° while that of the black superhydrophobic 2024 aluminum alloy surface prepared by the conventional one-step chemical etching method using NaOH solution as the etchant was 30.8°. The improved mechanical durability of the as-prepared non-black superhydrophobic 2024 aluminum alloy surface can be ascribed to the rough surface with relatively regular micro-scale dents and some nano-scale fibers in the micro-scale dents, which was confirmed by SEM observation.

**Author Contributions:** Conceptualization, methodology, L.-M.S. and G.-B.L.; software, H.T.; writing—original draft preparation, L.-M.S. and G.-B.L.; writing—review and editing, H.T. and Z.-H.L.; project administration, L.-M.S.; acquisition, J.-Y.W. All authors have read and agreed to the published version of the manuscript.

**Funding:** This work was supported by the Sichuan Science and Technology Program (No.: 2021YFG0283), the Deyang Science and Technology Program (No.: 2020SZZ046) and the Natural Science Foundation of Sichuan Engineering Technical College (No.: YJ2020KJ-07).

**Institutional Review Board Statement:** Not applicable.

**Informed Consent Statement:** Not applicable.

**Data Availability Statement:** The data that support the findings of this study are available from the corresponding authors, L.-M.S., upon reasonable request.

**Conflicts of Interest:** The authors declare no conflict of interest.

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
