# Peer review of "A Novel Simple Fabrication Method for Mechanically Robust Superhydrophobic 2024 Aluminum Alloy Surfaces"

_coatings, doi:10.3390/coatings12111717_

Round 1
Reviewer 1 Report
In this article, the authors describe the process of etching an aluminum alloy to give the surface a hierarchical structure necessary to achieve a superhydrophobic state. However, this material does not contain a fundamental novelty: etching with a NaOH solution was previously shown, the authors propose the second stage of processing (etching with a sodium carbonate solution, but they do not give a mechanism for the ongoing processes). Surface treatment with fatty acids is a well-worn topic in the scientific community and is also not scientifically new.
In this edition, I do not recommend the article for publication. In order to improve the article, it is necessary to rework the material and correct the following remarks:
1) The purpose of the work is written incorrectly (this applies more to the conclusions).
2) L97 "the passivation of low surface energy". It is not clear what the authors meant.
3) It is necessary to present the proposed mechanism of etching in sodium carbonate solution.
4) The microphotographs show that the surface is smoothed as a result of washing in hydrochloric acid. Also, as a result of etching, a regular structure is not formed, this is a random process. On fig. 2e, the surface structure repeats the previous one, and at a higher magnification it is not possible to consider a new level of roughness. It is likely that the SEM micrographs need to be altered or edited to show the structures formed and their sizes.
Figure 3a also shows that the surface looks smooth.
5) Table 1. It is necessary to give the Al content in percent. Why is there practically no oxygen on the surface of the original aluminum? Why is there so much copper and oxygen on the surface as a result of etching in NaOH solution? Are copper oxyforms formed that are not washed from the surface?
Point d: Why does the Si content increase as a result of modification with stearic acid?
6) Authors should provide data on contact angles on all types of surfaces: before and after modification. Angles of 160 degrees on a smooth surface raise questions. Provide photos or videos.
7) Figure 4c shows a high intensity of Al2O3 for untreated aluminum, which is inconsistent with Table 1.
8) Fig.6. It is necessary to provide: a) a photo of the surface after etching in sodium carbonate without StA; b) photo after etching in NaOH at 90C after StA modification; c) Why is the sample black? Etching products are not washed? d) the authors write that samples a and b are silvery, but they are white in the photo, which indicates an excess of StA on the surface.
9) The abrasion resistance test carried out is not informative, because does not reflect the dynamics of contact angle change. The properties deteriorate significantly and all samples do not show mechanical resistance. It is necessary to measure the contact angle depending on the amount of abrasion path (for example, every cm). Also, for the contact angles, the confidence interval must be given.
Author Response
Thank you for your suggestions. We have revised manuscript based on your suggestions. The major revised portions were marked in red for easier review purpose. If you have any question about the revised manuscript, please don’t hesitate to let us know. Please see the attachment

Reviewer 2 Report
Title: A novel simple fabrication method for mechanically robust superhydrophobic aluminum alloy surfaces
Comments:
Update the literature review
Define the research gap
State specific contributions at the end on introduction
Why two-step chemical etching method
Modify figure 1 and figure 2. There is a red line below the words
The authors stated, “The prepared superhydrophobic surface using the two-step chemical etching method shows an excellent superhydrophobic performance and a desirable mechanical durability.”. This needs more justification.
The results need to discuss deeply
Please add new separate section for discussion
Please compare your results with published ones
Author Response
Thank you for your suggestions. We have revised manuscript based on your suggestions. The major revised portions were marked in red for easier review purpose. If you have any question about the revised manuscript, please don’t hesitate to let us know. Please see the attachment.

Reviewer 3 Report
Journal: Coatings.
Manuscript ID:
Title: Type of manuscript: Article
Title: A novel simple fabrication method for mechanically robust superhydrophobic aluminum alloy surfaces (Li-Mei Shan, Guo-Biao Liu, Hua Tang, Zhi-Hong Li and Ju-Ying Wu).
Manuscript details:
Journal: Coatings
Manuscript ID:
Type of manuscript: Article
Title: A novel simple fabrication method for mechanically robust superhydrophobic aluminum alloy surfaces
Rate the Manuscript:
Significance to field and specialization of “Coatings” journal: good.
In the paper it has been established that the mechanical durability of superhydrophobic aluminum alloy surface is an important indicator to the practical use. A great improvement on the mechanical durability of super hydrophobic aluminum alloy surface has been achieved by a two-step chemical etching method using NaOH solution as etchant and Na2CO3 solution as etchant, respectively. The robust mechanical durability can be attributed to a hierarchical rough surface with regular micro-scale dents and some nano-scale fibers in micro-scale dents.
The main conclusions:
The as-prepared aluminum alloy superhydrophobic surface with some regular micro-scale dents and some nano-scale fibers in micro-scale dents presents a robust mechanical durability which is confirmed by SEM observation and the smaller variation of water contact angle obtained from the samples before and after an abrasion test.
Question: Haw about the implementation of the current results to obtained mechanical properties?
The as-prepared superhydrophobic aluminum alloy surface retains a silvery color instead of black one of superhydrophobic aluminum alloy surface prepared by the conventional one-step chemical etching method using NaOH solution as etchant.
Question: please compare with other type of aluminum alloy surface?
The robust mechanical durability is confirmed by the big water contact angles and an excellent corrosion resistance exhibited after an abrasion test. The water contact angles of the as-prepared superhydrophobic aluminum alloy surface before and after an abrasion test are 161.7° and 144.6° while those of superhydrophobic aluminum alloy surface prepared by a conventional one-step chemical etching method using NaOH solution as etchant are 164.7° and 133.9°.
Questions: What is the main question addressed by the research?
The references are appropriate.This research based on over 65 published scientific works. What specific improvements should the authors consider regarding the methodology and treatment? (for example: Laser treatment of plasma coatings // Soviet Materials Science. - 1991, vol.27, No 1.- P.51-55. https://doi.org/10.1007/BF00724136).
Scientific content: good.
Originality: good.
Clarity and presentation: acceptable.
Appropriateness for Journal: appropriate subject mater for the “Coatings”
Need for rapid publication: no.
Conclusions consistent with the evidence and arguments
presented and they address the main question posed and practically similar to abstract.

Author Response

(The authors gave the same response as above.)

Reviewer 4 Report
1. Author must revise the manuscript for grammatical and typographical errors.
2. It is recommended to format the manuscript as per the journal’s requirement.
3. Spacing from line 26 to 34 seems different than the rest of the manuscript. Please revise it.
4. What is the reason for many elements like Cu, Mg, Fe, Mn, and Si are present on the surface as reported in elemental mapping (figure 3)?
5. It is recommended to add the composition of Aluminum substrate in the beginning of the results to avoid any confusion to the readers.
6. Resolution of figure 4 must be improved.
7. Line 267 have different font sizes, Revise it.
Author Response

(The authors gave the same response as above.)

Round 2
Reviewer 2 Report
The paper can be accepted
Author Response
Thank you for your suggestions. 1) We have added some descriptions about the mechanical durability of superhydrophobic aluminum alloy surfaces mentioned in previous literatures [5–15]. Especially, a fully description about the mechanical durability of superhydrophobic aluminum alloy surface prepared by the NaOH solution etched mentioned in previous literature [15]. 2) Based on the revisions in the introduction, we have re-arranged the references. 3) We have added some discussion for Figures 2 and 6. 4) We have checked English language and style and corrected some mistakes.